# Cellular and Molecular Determinants of Biologic Drugs Resistance and Therapeutic Failure in Inflammatory Bowel Disease

**DOI:** 10.3390/ijms25052789

**Published:** 2024-02-28

**Authors:** Pierluigi Puca, Ivan Capobianco, Gaetano Coppola, Federica Di Vincenzo, Valentina Trapani, Valentina Petito, Lucrezia Laterza, Daniela Pugliese, Loris Riccardo Lopetuso, Franco Scaldaferri

**Affiliations:** 1Dipartimento di Medicina e Chirurgia Traslazionale, Università Cattolica del Sacro Cuore, 00168 Rome, Italy; pgpuca@gmail.com (P.P.); capobianco_ivan@yahoo.it (I.C.); federica.divincenzo30@gmail.com (F.D.V.); 2IBD Unit, UOC CEMAD Centro Malattie dell’Apparato Digerente, Dipartimento di Scienze Mediche e Chirurgiche Addominali ed Endocrino Metaboliche, Fondazione Policlinico Universitario Agostino Gemelli IRCCS, 00168 Rome, Italy; gcopp@gmail.com (G.C.); valentina.petito@unicatt.it (V.P.); lucrezia.laterza@policlinicogemelli.it (L.L.); daniela.pugliese@policlinicogemelli.it (D.P.); lopetusoloris@gmail.com (L.R.L.); 3Alleanza Contro il Cancro, Istituto Superiore di Sanità, 00144 Rome, Italy; valentina.trapani@unicatt.it; 4Department of Medicine and Ageing Sciences, “G. d’Annunzio” University of Chieti-Pescara, 66100 Chieti, Italy; 5Center for Advanced Studies and Technology (CAST), “G. d’Annunzio” University of Chieti-Pescara, 66100 Chieti, Italy

**Keywords:** drug resistance, therapeutic failure, non-response, biologic therapy, inflammatory bowel disease

## Abstract

The advent of biologic drugs has revolutionized the treatment of Inflammatory Bowel Disease, increasing rates of response and mucosal healing in comparison to conventional therapies by allowing the treatment of corticosteroid-refractory cases and reducing corticosteroid-related side effects. However, biologic therapies (anti-TNFα inhibitors, anti-α4β7 integrin and anti-IL12/23) are still burdened by rates of response that hover around 40% (in biologic-naïve patients) or lower (for biologic-experienced patients). Moreover, knowledge of the mechanisms underlying drug resistance or loss of response is still scarce. Several cellular and molecular determinants are implied in therapeutic failure; genetic predispositions, in the form of single nucleotide polymorphisms in the sequence of cytokines or Human Leukocyte Antigen, or an altered expression of cytokines and other molecules involved in the inflammation cascade, play the most important role. Accessory mechanisms include gut microbiota dysregulation. In this narrative review of the current and most recent literature, we shed light on the mentioned determinants of therapeutic failure in order to pave the way for a more personalized approach that could help avoid unnecessary treatments and toxicities.

## 1. Introduction

Inflammatory Bowel Disease (IBD), including as two primary manifestations Crohn’s disease (CD) and ulcerative colitis (UC), stands as a formidable challenge within the spectrum of gastroenterological disorders. These chronic inflammatory conditions are characterized by unpredictable and recurrent flare-ups in the context of chronic inflammation, which significantly compromise the quality of life for the affected individuals. As the global prevalence of IBD continues to rise, reaching an estimated 6.8 million individuals worldwide, the necessity of enhancing the therapeutic strategies becomes increasingly evident [1].

The available therapeutic options for IBD have evolved significantly over the years, with a remarkable emphasis on biologic therapies. While conventional treatments such as corticosteroids, mesalamine, and immunomodulators continue to play a role in managing symptoms and in treating mild to moderate disease, biologics drugs have emerged as pivotal agents, revolutionizing the treatment landscape of moderate to severe, corticosteroid-refractory disease. In particular, over the last couple of decades, the anti-α4β7 integrin antibody vedolizumab and the anti-Il12-23 ustekinumab have flanked antitumor Necrosis Factor α (anti-TNFα) agents (infliximab, adalimumab, and golimumab) in the treatment of both UC and CD. Furthermore, a new class of drugs known as “small molecules” (tofacitinib, upadacitinib, and filgotinib), mainly targeting JAK pathways, have been added to the therapeutic landscape of UC, as well as new and more innovative biologics or molecules that are being constantly approved and launched for both the afflictions [2].

However, despite the undeniable success of these therapeutic interventions, a substantial subset of patients still experiences therapeutic failure in the form of either primary non-response or secondary loss of response. The implications of therapeutic failure extend beyond the immediate clinical ramifications, encompassing economic burdens, compromised quality of life, and the need for surgical interventions in extreme cases.

The mechanisms underlying therapeutic failure are multifaceted, involving cellular, molecular, and microbiological factors that elude a one-size-fits-all explanation.

Despite the implications of therapeutic failure, a critical gap in our comprehension of the underlying mechanisms remains. This knowledge deficit jeopardizes the development of targeted interventions to overcome resistance, leaving clinicians with limited options and a lack of elements to predict response when faced with patients who exhibit suboptimal effectiveness. Therefore, a comprehensive exploration of the cellular and molecular pathways that govern drug resistance and therapeutic failure in IBD is not only advisable but urgent.

This narrative review embraces a journey to unravel the enigma of drug resistance in IBD, with a particular focus on the cellular and molecular dimensions according to the most recent scientific evidence. By delving into the intricacies of these mechanisms, we aim to pave the way towards personalized and effective therapeutic strategies, ultimately ushering in a new era in the management of IBD.

## 2. Materials and Methods

A bibliographic search was performed using the electronic databases PubMed, Scopus, and Embase. The search terms “biologics”, OR “biological drug”, OR “biological medication”, OR “small molecules” were matched with the words “IBD”, OR “Inflammatory bowel disease”, OR “Ulcerative colitis”, OR “Crohn’s Disease” and with the words “outcome”, OR “mechanism of resistance”, OR “failure”. All the terms were searched both as keywords and Medical Subject Headings (MeSH). No language restriction was used in the search filter. We hand-searched the bibliographies of relevant (according to their titles and abstracts) articles to provide additional references. The last search was run on the 31 December 2023. All the authors reviewed the titles and abstracts of English language articles individually to determine the relevance of the study. When this was not apparent, the full text of the articles was retrieved and reviewed. The results of the selected articles were compared to ultimately include only the most relevant ones. Comments, letters, and opinions were not considered in the search.

## 3. Genetic Polymorphisms

Genetics stand at the forefront of unraveling the intricate tapestry of IBD, shaping its pathogenesis, influencing prognosis, and serving as a key determinant in predicting therapeutic responses. The pursuit of understanding the genetic underpinnings of IBD has led to the identification of specific polymorphisms associated with differential drug responses, providing the basics for personalized treatment strategies [3]. These include Human Leukocyte Antigen (HLA) variants and polymorphisms in various other genes. These genetic factors not only affect disease onset and severity but also play a critical role in the variable response patterns observed in individuals subjected to therapeutic interventions.


**Human Leukocyte Antigens variants**


Human Leukocyte Antigens (HLAs) constitute a fundamental component of the immune system, playing a pivotal role in immune surveillance and response. HLAs are cell surface proteins, encoded by genes within the major histocompatibility complex (MHC), and serve as crucial mediators in presenting antigens to T cells. HLA molecules are integral to the recognition of self from non-self, governing the orchestration of immune responses and contributing significantly to immune-related pathologies. The role of HLA variants is gaining rising attention as a pathogenetic and prognostic determinant in several fields of medicine, especially in the field of immune disorders [4,5].

The HLA-DQA1*05 variant (rs2097432) has been extensively investigated, considering its alleged immunogenicity to tumor necrosis factor antagonists with subsequent resistance to therapies. The speculation derives from recent trials and genome-wide association studies (GWAS) that have associated the variant with increased anti-drugs antibody formation [6]. This hypothesis has been strengthened by evidence that the effect was less evident in patients under immunosuppressant therapy [7] and has led to the development of a rapid pharmagenomics assay, which has been validated in a recent study [8]. However, these data were not confirmed in real life cohorts and retrospective studies, for either anti-TNFα [9,10], or vedolizumab and ustekinumab [11]. In a recent Spanish investigation, HLA-DQA1*05 was associated with secondary loss of response only in patients receiving adalimumab without further immunosuppression, while, for patients under other biologic treatments (encompassing infliximab, vedolizumab, and ustekinumab), no association was detected [10]. Furthermore, a post-hoc analysis conducted on patients from IM-UNITI (CD) and UNIFI (UC) failed to establish a relationship between serum drug concentrations of ustekinumab, loss of response, and anti-drug antibodies formation [12].

Other HLA variants have been associated with lack of response to infliximab, including the SNP rs2395185, an intronic variant in HLA-DRB9. In a GWAS dating back to 2010, this SNP was associated with a primary lack of response to infliximab in a pediatric population [13]. The long-term role of both rs2097432 and rs2395185 HLA variants was for the first time assessed by a very recent, multicentric, ambispective study in a large pediatric cohort of IBD patients. Both variants were associated with lack of response to infliximab after a follow up period ranging from 3 to 9 years [14].

Last but not least, HLA-DRB1 alleles could play a role in determining immunogenicity to infliximab in IBD, especially with the presence of arginine at position 74, as well as the absence of glutamate at position 71, in the peptide-ligating site of the HLA-DRB1 [15]. The immunogenicity towards TNFα inhibitors agents has been demonstrated with adalimumab as well, although in the context of other autoimmune disorders such as rheumatoid arthritis and hidradenitis suppurativa [16].

The molecular and cellular mechanisms underlying the association between HLA polymorphisms and therapeutic resistance involve the role of HLA molecules in antigen presentation and immune response. HLA genes encode cell surface proteins expressed by antigen processing cells that present antigens to T cells, thereby initiating an adaptive immune response towards specific antigens, usually belonging to the non-self. Specific HLA variants can present drug-related antigens to T cells, leading to the production of antibodies against the drug. The underlying mechanism may involve molecular mimicry, whereby the immune system recognizes a drug as a foreign antigen due to its similarity to self-antigens.


**Other Polymorphisms**


While classic and pivotal pharmacogenetic studies used to focus only on single genes or groups of correlated genes, modern pharmacogenomics prefers a wider approach. For example, a systematic review from Bek and colleagues associated a series of variants with response to infliximab in CD and UC, although with weak statistical outcomes. Such variants were related to genes involved in innate immune response, such as the recognition of bacterial components (TLR2, TLR4, and TLR9) and cytokine pathways (TNFRSF1A, IFNG, IL6, and IL1B) [17]. On the contrary, pharmacogenetic loci associated with primary non-response to anti-TNFα agents emerged from an unbiased GWAS in pediatric IBD patients by Dubinski et al. In particular, three loci were significant in a final predictive model: TACR1 (Tachykanin Receptor 1), a receptor for substance P, which is a known pro-inflammatory molecule; PHACTR3 (Phosphatase And Actin Regulator 3), which is associated with the nuclear scaffold in proliferating cells; and FAM19A4 (Family With Sequence Similarity 19 Member A4, C-C Motif Chemokine Like), which functions as a chemokine and regulator of immune cells in the brain [13]. A systematic review by Lauro and colleagues identified three big groups of polymorphisms that interfere with different mechanisms with response to anti-TNFα biologics [18]:-Polymorphisms of the TNFα gene (leading to increased cytokine secretion) and TNFα receptors genes (TNFR1/2, resulting in increased response after interaction with TNFα).-Polymorphisms of innate immunity-related genes (TLR4, CD14, IL-6, and IL-1β).-Polymorphisms of apoptosis- and autophagy-related genes (FASL, CASP9 and ATG16L1), probably through an inhibitory effect on the apoptosis of immune cells.

Among the genes involved in the apoptotic process, CASP9 has given the most convincing evidence. A recent study investigated CASP9 variants both in mucosal biopsies and peripheral blood white cells, discovering a close relationship between the variants rs1052571 and rs4645978 and patients’ response to infliximab in CD [19].

Little and uncertain information is known about polymorphisms that could be related to resistance or response to other biologic drugs. The rs7234029 polymorphism in the Protein Tyrosine Phosphatase Non-Receptor Type 2 (PTPN2, a protein involved in chronic inflammation) could reduce the response to ustekinumab in patients with CD, although this result emerged from a monocentric, retrospective uncontrolled trial [20].

Interestingly, NOD2/CARD15 has been the first and main gene with a proven association with IBD susceptibility. Polymorphisms in this genetic sequence have been investigated as possible determinants of resistance to infliximab and other biologic drugs, with consistent negative results throughout the years both for patients with CD and with UC [21,22].

The multidrug resistance 1 (MDR1) gene, also known as ABCB1, encodes a membrane-bound efflux transporter, P-glycoprotein, which is involved in the transport of a wide range of substrates, including drugs, across cell membranes. In the context of IBD, MDR1 has been a subject of interest due to its potential implications in disease pathogenesis and treatment response. While its physiological function in the gut is not fully understood, its high levels of expression suggest a role in protection against xenobiotics. Historical studies have explored the association between MDR1 gene polymorphisms and IBD, indicating a potential link between genetic variations in MDR1 and susceptibility to IBD, as well as its influence on drug response and resistance [23].

The first study associating MDR1 polymorphisms with refractory IBD dates to 2004. Potocnik and colleagues showed that SNPs in MDR1 that were functionally correlated with the overexpression of P-glycoprotein were associated with CD being refractory to conventional therapies. Indeed, a suggested role for P-gp is glucocorticoid transport and protection against glucocorticoid-induced apoptosis of T-lymphocytes [24]. However, polymorphisms of MDR1 and other ATP-binding cassettes transporters did not show a correlation with lack of response to infliximab in a Hungarian cohort of IBD patients [25].

Although further data highlighted the relationship between certain MDR1 SNPs and glucocorticoid resistance, no correlation between this gene and refractoriness to biologic drugs is currently known. Therefore, the mechanisms by which MDR1 variants determine resistance to conventional IBD drugs can possibly be ascribed to their function as xenobiotic (and drug) efflux pumps [26].

Genetic variability could also be a predisposing factor for lower serum drug levels, thus leading to therapeutic resistance and side effects.

Interestingly, Tang and colleagues found an association between specific SNPs in certain cytokines and infliximab trough levels after the induction phase. GA carriers of rs442905 within C1orf106 (an epithelial barrier regulator and innate immunity activator) showed lower infliximab levels in comparison to GG + AA carriers, and GG + AA carriers rs3213448 of IL1RN showed higher infliximab levels than GA carriers. Finally, GG carriers of rs7587051 within ATG16L1 (a regulator of autophagy) had lower drug levels than GC + CC carriers [27]. Similarly, in a pediatric cohort of IBD patients receiving either infliximab or adalimumab, the main polymorphisms associated with a suboptimal drug concentration of infliximab were in the TLR2 and LY96 genes; interestingly, LY96 is a protein that acts as an enhancer of the lipopolysaccharide-TLR interaction, thus amplifying inflammatory stimuli via NF-κB. However, suboptimal trough levels of adalimumab were associated with variants in TLR4 and TNFRSF1B (a transmembrane glycoprotein that can induce cell apoptosis and survival) [28]. A study from the same group on a similar subset of patients also attributed a role to the variant rs3024505 T of IL10 (with no meaning in gene transcription) in determining infra-therapeutic drug levels [29]. Table 1 summarizes relevant SNPs associated with an impact of therapeutic outcomes.

Among cytokines, interleukin-23 (IL-23) has often come under the spotlight as a possible determinant of drug resistance. IL-23 has been identified as a key driver of persistent intestinal inflammation in IBD. It is produced by macrophages and dendritic cells and exerts its effects via the IL-23 receptor (IL-23R) [30]. Many SNPs of IL-23R have been associated with susceptibility to IBD [31,32], and, recently, some genetic variants of IL-23R have been linked to the infliximab response rate in IBD patients. In fact, Jürgens et al. found that IL-23R SNPs (rs1004819, rs2201841, rs10889677, rs11209032, and rs1495965), which determine an increased risk of developing IBD, were also associated with an increased infliximab response rate compared to variants which determined a reduced risk of developing IBD (rs7517847, rs10489629, rs11465804, and rs1343151) (74.1% vs. 34.6% *p* = 0.001) [33]. Conversely, IL23R rs10489629-TT has been recently identified as a protective factor against infliximab failure in CD patients [34]. Furthermore, IL-23R polymorphisms have been related to paradoxical reactions associated with anti-TNFα therapy in IBD patients, leading to the need for drug discontinuation. Infliximab-induced psoriasis rates were higher in patients that were homozygous for some IL-23R genetic variants (rs10489628, rs10789229, and rs1343151), according to a retrospective study in a population of pediatric CD patients [35]. Cravo et al. found that, in carriers of rs1004819 and rs10889677 IL-23 alleles, susceptibility to extra intestinal manifestations was increased, although an increase in failure to infliximab in terms of gastrointestinal disease was not detected [36]. The rs10889677 IL-23 variant was also found to be associated with a risk of developing alopecia areata [37]. Tillack et al. investigated the role of the rare G/G variant rs11209026 (p.Arg381Gln) of IL23R in determining the onset of skin psoriasiform lesions in patients receiving anti-TNFα; although the association was not statistically significant, all of the patients with severe psoriasiform skin lesions and/or anti-TNF-induced alopecia switched to ustekinumab were carriers of this variant [38]. Conversely, the same IL-23 polymorphism was linked with statistical significance to an increased susceptibility to paradoxical psoriasiform reactions during anti-TNF therapy in patients with psoriasis or other autoimmune conditions [39]. Remarkably, the rs11209026 IL-23 variant has also been associated with a heightened risk of CD-related surgeries, suggesting a more complex role in determining disease phenotype beyond drug resistance [40]. Table 2 summarizes the available evidence on the role of IL23R polymorphisms in determining therapeutic outcomes.

## 4. Transcriptional Profiles

Studies investigating the transcriptional profiles associated with drug resistance in IBD have explored both combined transcriptional shifts and transcriptional shifts related to single genes. The identification of transcriptional signatures and regulatory networks may contribute to the development of precision medicine approaches to IBD, allowing for the assessment of disease course and outcomes, and the identification of novel therapeutic targets.

Oncostatin M (OSM) is a secreted cytokine that belongs to the interleukin-6 (IL-6) family of cytokines and can bind to two different receptors, the Leukemia inhibitory factor receptor (LIFR) and the OSM receptor (OSMR). It plays a multifaceted role in various physiological processes, including differentiation, cell proliferation, and immune regulation. OSM is involved in homeostasis and has been implicated in diseases characterized by chronic inflammation, such as rheumatoid arthritis, lung and skin inflammatory conditions, atherosclerosis, and cardiovascular disease. In the context of IBD, OSM has emerged as a biomarker of diagnosis, worse prognosis, and therapeutic non-response [41,42].

The pivotal study on the role of OSM was performed by West and colleagues, who showed that OSM is overexpressed in patients with IBD, particularly by those with deep ulcerations. At a molecular and cellular level, haematopoietically derived OSM interacts with stromal-expressed OSMR to trigger an inflammation cascade. The interaction with OSMR is complementary and synergistic with TNFα (and other cytokines belonging to the IL6 group), which explains the resistance to anti-TNFα therapy in patients overexpressing OSM [43]. This assumption has then received several confirmations throughout the years. On the one hand, elevated serum OSM has been undoubtedly related to poor clinical and endoscopic outcomes for patients receiving infliximab [44,45]. On the other hand, fecal OSM has been tested as a predictor of response to infliximab by Cao and colleagues with promising but still inconclusive results. In a population of both CD and UC, the role of fecal OSM taken alone is not very convincing in determining resistance to infliximab, but the clinical statistical significance is noticeably increased if OSM is coupled with fecal calprotectin [46].

As new biologic drugs have received approval for IBD, new studies concerning OSM as a predictor of response or failure are expected. So far, the only study assessing the role of OSM in determining resistance to vedolizumab did not find any significant relationship between serum OSM levels and mucosal healing in both UC and CD. On the contrary, statistical significance was detected in the anti-TNFα cohort of the same study, with low OSM blood levels predicting mucosal healing [47]. In the same way, in a study comparing two cohorts under therapy with infliximab and ustekinumab, mucosal OSM levels were associated with lack of response in the infliximab cohort but not in the ustekinumab one [48].

TREM-1 (Triggering Receptor Expressed on Myeloid Cells 1) is a cell surface receptor primarily expressed on neutrophils and monocytes. It plays a crucial role in amplifying the inflammatory response by triggering the production of pro-inflammatory cytokines and chemokines. In the context of health, TREM-1 is involved in the regulation of immune responses to infections. However, its dysregulation has been associated with various inflammatory and autoimmune diseases, including IBD. The overexpression of TREM-1 from myeloid cells (macrophages and neutrophils), together with low concentrations of this molecule in whole blood, were associated with lack of response to infliximab in a meta-analysis of 2019. The authors concluded that overexpression of TREM-1 generates infliximab resistance through increased TNFα production and downstream upregulation of CCL7, a promigratory cytokine that perpetuates inflammation [49]. These results, however, are not consistent with other and following findings. Vermiere and colleagues, for example, detected an opposite trend, with low TREM-1 serum (and mucosal) concentration predicting response to anti-TNFα but not to vedolizumab and ustekinumab [50]. No other studies have assessed the role of TREM-1 in resistance to vedolizumab or ustekinumab to date.

Among the most eligible candidates as markers of therapeutic response in IBD, attention has been focused on IL13RA2 (IL-13 Receptor alpha 2), whose predictiveness of non-response to infliximab had been investigated at the beginning of the previous decade, when an increased expression of this gene was detected in the mucosal specimens of IBD patients benefitting from infliximab therapy [51,52]. In more recent studies, these associations have been extended to other anti-TNFα agents as well; mucosal concentration of IL13RA2 predicts lack of response to adalimumab but not to vedolizumab [53]. Of note, an increased mucosal expression of IL13RA2 has also been associated with corticosteroid refractoriness with a consequent need for infliximab escalation [54]. Last but not least, a new study on patients with moderate to severe UC, analyzing mucosal gene expression through an Artificial Intelligence algorithm, confirmed the role of this gene in predicting failure of anti-TNFα treatment [55]. IL13RA2 is a decoy receptor for IL13 with a non-canonical JAK/STAT signaling activation. IL13RA2 binds IL13 with a stronger affinity than its physiological receptors, thus working as a physiological inhibitor of type-2 immunity by limiting the availability of free IL-13 that would otherwise drive STAT6-dependent signaling [56]. Further mechanistic insight was provided by Verstockt and colleagues, who showed that IL13RA2-deprived mice exposed to dextran sodium sulfate (DSS) have similar colitis severity but quicker recovery in comparison to healthy mice exposed to DSS. This phenomenon is caused by IL13RA2 expressed on epithelial cells that negatively affects goblet cells recovery [57].

As new biologic drugs become approved, new investigations dig deeper into transcriptional profiles related to therapeutic resistance. The mucosal baseline expression of four genes (PIWIL1, MAATS1, RGS13, and DCHS2) has been related and then validated as a marker of vedolizumab failure, while no correlation was detected for patients receiving infliximab. Interestingly, MAATS1 and RGS13 are expressed by endothelial cells and could play a role in cell migration and diapedesis, while DCHS2 is expressed by epithelial cells and probably plays a role in modulating innate and adaptive immunity [58]. Whole blood transcriptional profiles have been linked to vedolizumab response as well. According to Haglund and colleagues, upregulated pathways in responders are related to “innate immunity”, “phagocytic processes”, “cytoskeleton modulation”, “amino acid transport across the plasma membrane”, and “glycosaminoglucan metabolism”. Conversely, downregulated pathways were “mitochondrial processes” such as “mitochondrial translation” and “respiratory electron transport”, “processing of rRNA” and “processing of tRNA” [59].

Mitochondrial disfunction in colonic macrophages of both patients with UC and DSS murine models has also been associated with more severe disease refractoriness to infliximab. In particular, mitochondrial disfunction is witnessed in this cell lineage by the absence or reduced expression of MCJ, a natural inhibitor of the respiratory chain Complex I [60].

Single-cell transcriptomics. The advent of single-cell RNA sequencing (scRNA-seq) has revolutionized transcriptomics, offering unprecedented insights into cellular heterogeneity. Unlike traditional bulk RNA sequencing, scRNA-seq enables the examination of gene expression at the single-cell level. ScRNA seq has provided useful information also on the mechanisms of drug resistance.

Smillie et al. have shown that, in UC, the presence of inflammation-associated fibroblasts (IL-13Rα2+IL-11+) was associated with resistance to anti-TNF treatment. Furthermore, in the same study they demonstrated that OSM phenocopies TNFα, thus providing further explanation of the TNF-inhibitors resistance in patients overexpressing OSM [61]. Conversely, Martin and colleagues investigated a cohort of CD patients and discovered a cellular signature consisting of IgG plasma cells, inflammatory mononuclear phagocytes, activated T cells, and stromal cells (renamed GIMATS) that is associated with reduced likelihood of response to anti-TNF treatment. Taken altogether, the GIMATS module promotes local recruitment, activation, and expansion of T cells, as well as stromal cells activation and possibly fibrosis development [62]. The finding of activated fibroblasts in the setting of anti-TNFα-resistant IBD is consistent with the study of Friedrich and colleagues. In this study, the cellular subset of “activated fibroblasts” (presenting a specific transcriptional signature with an elevated expression of M4/M5 modules) attracts neutrophils via IL1β (but not TNFα)-driven interaction, thus explaining the resistance to anti-TNFα agents. Furthermore, the elevated expression of M4/M5 transcriptional modules was found to be associated with deep ulcerations and an elevated Nancy Index on histology [63].

Single-cell transcriptomics also allows, thanks to the application of complex algorithms, the determination of a special characterization of cell subtypes. Interestingly, it has been shown that therapy with TNF inhibitors mainly reduces T cells and B cells compartments, and this phenomenon is more evident in women than men, thus providing a possible explanation for the higher rates of response observed in women [64].

Single-cell transcriptomics has found an application in pouchitis as well. The role of IL1B in determining resistance to therapies has been confirmed for vedolizumab. In fact, an increased interaction between IL1B+/LYZ+ monocytes/macrophages (phagocytic cells with a strong antimicrobial activation) and TH17 polarized T cells has been detected in patients with pouchitis that is refractory to vedolizumab therapy [65]. Figure 1 shows the mechanisms determining drug resistance according to studies of single-cell transcriptomics. 

## 5. Epigenetic Modifications

DNA methylation signatures. In the era of biobanks and prospective datasets, epigenetics represents an emerging side of multi-omics to aid the identification of predictive biomarkers and precision medicine in IBD [66].

Gene expression can be extensively modulated by epigenetic changes, which can lead to both gene enhancement and silencing, alternative splicing, and several other transcriptional modifications [67]. Since evidence of specific methylation profiles associated with both IBD subtypes and activity is available [68,69], preliminary data on therapy response have also been generated, despite the burden of dynamic confounding factors, the high variability of methods, and the challenging implementation in clinical practice [70].

To this purpose, Lin et al. recently carried out a bold study on epigenomics using the data collected from the prospective, multicentric cohort of the “Personalized Anti-TNF Therapy in Crohn’s disease (PANTS)” study; the PANTS study had previously revealed that, among many clinical and biochemical factors, the only independent factor associated with primary clinical non-response to anti-TNFα therapy in CD patients was a low drug concentration at week 14 [71]. Consequently, the authors analyzed the DNA methylation profiles from 1104 whole blood samples of 385 patients collected at baseline, week 14, 30, and 54, to identify epigenetic signatures that could predict low drug concentration, thus being possibly associated with a primary non-response to anti-TNF therapy. Interestingly, 323 differently methylated positions (DMPs) at baseline were associated with higher anti-TNF drug concentrations at week 14. Among these DMPs:-There were 26 associated with specific immune compartment cells (B cells, T cells, granulocytes, and monocytes).-A further 125 had been previously associated by epigenome-wide association studies (EWAS) with alcohol consumption, body mass index, smoking, C reactive protein, and IBD type [72].

According to another retrospective evaluation of patients of the PANTS study, baseline expression of major histocompatibility complex, antigen presentation, myeloid cell enriched receptor, and other innate immune gene modules were significantly higher in anti-TNFα (infliximab and adalimumab) responders in comparison to non-responders [73]. Data coming from the rheumatoid arthritis world has detected differences in the methylation profile of T cell activation and differentiation, GTPase-mediated signaling, and actin filament organization pathways between responders and non-responders to infliximab [74].

Similar studies also allow the delineation of a temporal characterization of the impact of biologic drugs on epigenetics. Mishra and colleagues prospectively collected multi-omics data from 14 IBD patients at seven time points from induction to week 14. Notably, the authors observed early profound changes (mostly in downregulating gene expression) already 4 h after the first exposure to infliximab, with a higher homogeneity and perdurance in gene expression in remitters compared to non-remitters during the study period at almost all the time points. Conversely, upregulated transcripts associated with therapeutic failure included TH2-related and eosinophil-related genes encompassing ALOX15, FCER1A, and OLIG2, as well as modules reflecting processes such as interferon signaling, erythropoiesis, and platelet aggregation [75].

MicroRNAs. MicroRNAs (miRNAs) play a supporting role in the multifaceted pathogenesis of IBD, mainly acting as regulators of gene expression. These small, non-coding RNA molecules exert significant influence over various cellular processes. In the context of IBD, dysregulation of miRNA expression has been implicated in the initiation and perpetuation of intestinal inflammation, although little information is available on the role of these molecules in determining resistance or response to medications. MiRNAs orchestrate a complex interplay by targeting the key genes involved in immune modulation, gut microbial homeostasis, epithelial barrier function, and cytokine signaling [76]. This nuanced regulatory network positions miRNAs as promising candidates not only in understanding the pathogenesis of IBD but also in unraveling their potential role in the development of resistance to therapeutic interventions [77].

Batra et al. have identified eight serum and biopsy miRNAs associated with better response to infliximab and corticosteroids (miR-126, miR-146a, miR-146b, miR-26a, miR-26b, miR-320a, miR454, and let-7c). In particular, miR-146a and miR-146b, overexpressed in inflamed tissues, are negative regulators of innate immune signaling, acting as an inhibitor of NF-κB; conversely, miR-320a is overexpressed in non-inflamed mucosa and works as a wound healing regulator. The other mentioned miRNAs play their role in M2 (anti-inflammatory) macrophages polarization (let-7c) and leukocyte trafficking (miR-126) [78]. Furthermore, according to very recent evidence, combo therapy with infliximab and granulocyte apheresis is able to modify the concentration of specific miRNAs, and some of them are associated with response or refractoriness to therapy [79]. However, it has to be signaled that the other studies failed to highlight a correspondence between miRNAs and response to therapies [80].

In any case, it has been suggested and demonstrated that infliximab therapy affects and modifies the expression of miRNA. For example, serum and fecal miR-126 and miR-20a, involved in immune regulation and epithelial barrier function, were significantly down-regulated after infliximab treatment in a pediatric population of CD patients [81]. At present, evidence on this subject appears inconclusive.

## 6. Other Mechanisms

Gut microbiota. Although an extensive discussion of the role of gut microbiota in determining therapeutic failure is beyond the scope of this paper, there are microbial-driven molecular and biological mechanisms that deserve to be briefly explained.

Heightened dysbiosis, elevated pro-inflammatory markers (especially IL-1β, IL-6, IL-17a, and TNF-α), and decreased levels of short-chain fatty acids (SCFAs) at the beginning of treatment have been associated with reduced response to infliximab and anti-TNFα agents. Dysbiosis, characterized by an imbalance in microbial composition, may escalate inflammatory cascades, fostering an environment less responsive to infliximab. Additionally, reduced SCFA levels, known for their anti-inflammatory properties, may compromise the regulatory balance within the gut, influencing immune responses and impacting the efficacy of infliximab [82,83,84].

Gut microbiota composition and function has also been associated with response or lack of response to vedolizumab. Levels of *Roseburia inulinivorans* (*Firmicutes*) have been directly correlated with response in CD patients receiving vedolizumab by Ananthakrishnan and colleagues. In fact, this species is responsible for SCFA production; furthermore, *Roseburia inulinivorans* encodes genes for flagellin proteins that determine an IL8-driven inflammation setting. Interestingly, according to the same authors, baseline-specific bacterial SNPs were associated with increased response to vedolizumab in CD (L-arginine biosynthesis) and UC (uridine monophosphatate biosynthesis and pentose phosphate pathway) [85].

The role of SCFA in determining disease severity and response to therapies is confirmed in anti-TNFα-refractory patients receiving ustekinumab. It has been ascertained that CD patients that are refractory to ustekinumab have lower baseline levels of *Faecalibacterium* (*Firmicutes*), one of the most important SCFA-producing bacteria [86].

Another possible microbiota-driven mechanism of drug resistance, shared by anti-TNFα and anti-α4β7 antibodies, could be the expression of IgG-degrading enzymes. This phenomenon was first described for *Streptococcus pyogenes* pathogens. In any case, a clear explanation has not been given on how intravenously administered antibodies can come into contact with luminal microbes [87,88,89]. A “leak” from the site of inflammation and immune cells recruitment towards the gut lumen could be hypothesized.

Diet, nutrition and obesity. The influence of nutrition biomarkers on the mechanisms of drug failure in the context of inflammatory bowel disease IBD is an area of emerging interest. For example, a very recent prospective observational study found that CD patients with a lack of response to anti-TNFα agents tended to have a lower intake of Zinc and Calcium in comparison to responders, although no significant differences were found in the intake of macronutrients [90].

The immunomodulatory effects of vitamin D on the gut mucosa and its role in regulating immune response, microbial homeostasis, and the epithelial barrier are well known. Vitamin D deficiency has been associated with a differential response to biologic agents in IBD, although with inconsistent findings. While recent data contrast as to the impact of serum vitamin D levels on the outcome of anti-TNFα therapy, a recent investigation from Abraham and colleagues has associated lower baseline vitamin D levels with reduced response to vedolizumab [91,92,93].

IBDs are increasingly considered Western diseases; therefore, besides the impact of nutritional deficiencies on biologic treatment, recent evidence has shed light on the role of obesity and hyper nutrition in determining reduced response to biologic drugs. The negative impact of obesity on treatment outcomes has been convincingly demonstrated for anti-TNFα agents (especially when administered subcutaneously), whilst conflicting evidence is available for vedolizumab [94,95]. The explanation between obesity and reduced response to therapies can be considered in relation to several mechanisms. Altered gut microbiota composition, which is known to occur in response to obesity, can impact the metabolism of various substances, including triacylglycerol and cholesterol. These alterations may in turn regulate adipogenesis and affect immune and inflammatory responses. Additionally, obesity is linked to dysregulation of adipocyte function and micro-environmental inflammatory processes, which can significantly influence insulin signaling and immune function. Furthermore, obesity-related dysregulation of adipokines and sirtuins may also play a role in modulating immune responses and inflammatory processes [96].

## 7. Conclusions, Limitations and Future Directions

Our exploration of the cellular and molecular determinants of biologic drug resistance in IBD highlights the complexity of therapeutic challenges. If genetic polymorphisms, particularly HLA variants and other polymorphisms, provide nuanced and sometimes inconsistent associations with treatment outcomes, transcriptomics and the advent of single-cell RNA sequencing and metagenomics guarantees a more detailed and exhaustive explanation of therapeutic resistance. Other aspects, including gut microbiota and miRNA, need to be considered as well.

The investigation of the cellular and molecular processes which determine drug resistance is not free of limitations and biases. In fact, studies investigating such pathways often present heterogeneity in patient recruitment and low attention to disease stratification. Moreover, the application of our knowledge in clinical practice is still jeopardized by the logic of cost-effectiveness, and studies are required to figure out which molecular markers of therapeutic failure are usable in daily clinical practice without excessive costs.

Looking forward, the complexity of these determinants necessitates a holistic approach. While individual elements may not exhibit strong statistical associations with therapeutic outcomes, the integration of genetic, transcriptional, and metagenomic data holds immense potential. Furthermore, a deeper understanding of the biological mechanisms underpinning therapeutic failure is of paramount importance to establish innovative and precise combination therapies of biologic drugs. The enhancement of combined therapies, in fact, would allow a larger coverage of drug resistance mechanisms. To this purpose, evidence has already been produced over the last few years on the use of biologic regimens combining ustekinumab and vedolizumab, even if data concerning the use of anti-TNFα inhibitors in combination with other agents are available as well [97]. Last but not least, strengthening our molecular knowledge of therapeutic failure is pivotal to discover novel therapeutic targets for future medications.

Collaboration among geneticists, immunologists, and gastroenterologists is pivotal for translating these findings into clinical applications. This collaborative effort is integral to establishing a paradigm shift toward personalized, patient-centered care.

## Figures and Tables

**Figure 1 ijms-25-02789-f001:**
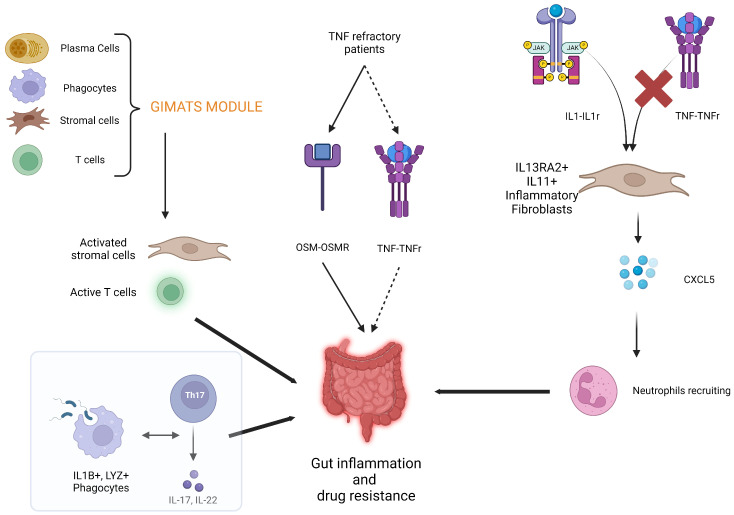
Mechanisms of biologic drug resistance unveiled by single-cell transcriptomics. Extensive explanation provided in the text.

**Table 1 ijms-25-02789-t001:** Relevant Single Nucleotide Polymorphisms with therapeutic influence.

SNPs	Molecule	Disease	Drug	Effect	References
rs2097432	HLA-DQA1*05	P-UC P-CD UC CD	IFX ADA	Decreased response	[10,14]
rs2395185	HLA-DRB9	P-UC P-CD	IFX	Decreased response	[13,14]
rs1052571 rs4645978	CASP9	CD	IFX ADA	Increased response	[19]
rs7234029	PTPN2	CD	UST	Decreased response	[20]
rs442905	C1orf106	CD	IFX	Decreased IFX levels	[27]
rs7587051	ATG16L1	CD	IFX	Decreased IFX levels	[27]
rs3213448	IL1RN	CD	IFX	Increased IFX levels	[27]
rs5030728	TLR4	P-UC P-CD	IFX	Decreased IFX levels	[28]
rs11465996	LY96	P-UC P-CD	IFX	Decreased IFX levels	[28]
rs1816702	TLR2	P-UC P-CD	ADA	Decreased ADA levels	[28]
rs3397	TNFRSF1B	P-UC P-CD	ADA	Decreased ADA levels	[28]
rs3024505	IL-10	CD	IFX	Decreased IFX levels	[29]

SNPs: Single Nucleotide Polymorphisms; UC: Ulcerative Colitis; P-UC: Pediatric Ulcerative Colitis CD: Crohn’s Disease; P-CD: Pediatric Crohn’s Disease; IFX: Infliximab; ADA: Adalimumab; UST: Ustekinumab.

**Table 2 ijms-25-02789-t002:** Relevant Single Nucleotide Polymorphisms in IL23R with therapeutic influence.

SNPs	Molecule	Disease	Drug	Effect	References
rs1004819 rs2201841 rs10889677 rs11209032 rs1495965	IL-23R	UC	IFX	Increased response	[33]
rs7517847 rs10489629 rs11465804 rs1343151	IL-23R	UC	IFX	Decreased response	[33]
rs10489629	IL-23R	CD	IFX	Increased response	[34]
rs10489628 rs10789229 rs1343151	IL-23R	P-CD	IFX	Increased IFX-induced psoriasis rates	[35]
rs1004819 rs10889677	IL-23R	UC	IFX	Increased EIMs rates	[36]
rs11209026	IL-23R	UC, CD, Psoriasis	IFX, ADA	Increased Anti-TNF-induced psoriasiform skin lesion rates; Increased risk of CD-related surgery	[38,39,40]

SNPs: Single Nucleotide Polymorphisms; UC: Ulcerative Colitis; CD: Crohn’s Disease; P-CD: Pediatric Crohn’s Disease; IFX: Infliximab; ADA: Adalimumab; AZT: Azathioprine; EIMs: Extra Intestinal Manifestations.

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
