# Peer review of "Cellular and Molecular Determinants of Biologic Drugs Resistance and Therapeutic Failure in Inflammatory Bowel Disease"

_ijms, 2024, doi:10.3390/ijms25052789_

Round 1

Reviewer 1 Report

Comments and Suggestions for Authors

Available therapeutic options for IBD have evolved significantly over the years, such as biologic therapies. Indeed, genetic polymorphisms, particularly HLA variations and other polymorphisms, provide nuanced and sometimes inconsistent associations with treatment outcomes. The topic is interesting., and only minor changes are required before publication.

Typeset in journal format.

What about nutritional therapy? I think the authors should discuss in the manuscript.

The mechanism of determinants of biologic therapies are not detailed enough.

Author Response

Thank you for the appreciation towards our job and for your suggestions.

Typesetting has been extensively improved, and modifications to layout, order or paragraphs references has been implemented.

Furthermore, in the "other mechanisms" paragagraph, a section discussing the role of diet, nutrition and obesity has been added. 

Reviewer 2 Report

Comments and Suggestions for Authors

This review manuscript titled "Cellular and molecular determinants of biologic drugs resistance and therapeutic failure in Inflammatory Bowel Disease" explores the multifaceted landscape of biologic drug resistance and therapeutic failure in Inflammatory Bowel Disease (IBD), encompassing Crohn's disease and ulcerative colitis. It examines various determinants including genetic polymorphisms, transcriptional profiles, microRNAs, and gut microbiota, shedding light on the complexities underlying treatment responses. Despite advancements in biologic therapies, a subset of patients experiences therapeutic failure, necessitating a deeper understanding of the molecular and cellular mechanisms involved. The review underscores the importance of interdisciplinary collaboration, personalized treatment strategies, and the exploration of novel therapeutic targets to optimize patient care and improve treatment outcomes in IBD.

 Some improvements can be made to improve this manuscript.

 Comments: 

  1. The whole paper’s structure needs to be reorganized. It seems like the authors compacted lots of information in this article without well-organization. For example, “DNA methylation signatures” part should not be included into “Chapter 3. Transcriptional profiles” because it is an epigenetic factor. “DNA methylation” and “MicroRNAs” could be combined as a new Chapter “Epigenetic effect”. The suggestion I mentioned is just an example and it is more than welcome if you have other ideas to organize the whole paper better.
  2. Another suggestion is to provide a clearer roadmap for readers to follow the complex molecular details. (A summarized graph for this review would be better).
  3. Providing more context about the limitations and future directions would be valuable. Acknowledging potential biases, targeting novel pathways, or combination therapies would be beneficial for readers.
  4. The whole paper’s information is dense and might be challenging for some readers to follow due to the complex molecular details and the extensive use of technical terms. Consider eliminating unnecessary technical details and focusing on the most critical information to improve overall readability.

Comments on the Quality of English Language

Extensive editing of English language required.

Author Response

Thank you very much for the appreciation of our paper and for the suggestions that we implemented in our text. 

First of all, extensive revision and editing of english language has been made. We hope that our paper is more readable now. 

Furthermore, we reorganized the paragraphs according to your suggestions, including micro RNAs and methylation signatures under the paragraph called "epigenetic modifications".

The final paragraph ("Conclusions, limitations and future directions") has been enriched and a reference to potential biases, potential novel pathways, or combination therapies has been added. 

Lat but not least, we tried to make this paper more readable for non expert readers by simplyfing english language and by providing essential explanation of some molecules' functions.

Reviewer 3 Report

Comments and Suggestions for Authors

I reviewed the manuscript entitled Cellular and molecular determinants of biologic drugs resistance and therapeutic failure in Inflammatory Bowel Disease.

I agree to accept this manuscript, but there are a few issues that need to be revised before acceptance. 

1) Line 50, What is the full name of UC and CD? Please supplement them.

2) Line 85, HLA, The title should not be abbreviated or changed to Human Leukocyte Antigens (HLA).

3) Line 102, TNFα, Use the full name when first appearing.Line 212, IL23 shoulde change to IL-23.

4) Line 221, P indicates that italics should be used for statistics, and the entire text should be checked and modified.

5) Line 261 and 291, IBD is sufficient, there is no need to write the full name anymore.

6) Line 318, what is DSS? Write the full name.

7) The format of references needs to be modified according to the requirements of the journal. For example, in ref 4, 5 and 7, only one author was written before using et al., while in ref 6, all authors were listed. The publication month of all refs can be deleted.

8) Some references published twenty years ago, and the author needs to update them.

9) I did not see the author's method of searching for literature. This paragraph should be added, including which databases were searched, search years, keywords, etc.

Author Response

Thanks for the positive opinion given to our manuscript and for your corrections that we implemented in order to improve our paper. 

In particular: 

  • the full name of UC and CD has been provided in the main text, at the beginning of the "introduction" paragraph;
  • the title of the paragraph about HLA has been modified and written without abbreviations
  • the antiTNF abbreviation has been clarified at the first appearance in the text and IL23 has been modified into IL-23
  • all the "p" indicating a statistical significance have been written in italics
  • lines 261 and 291 have been modified accordingly
  • the abbreviation of DSS has been clarified
  • references have been written with a software package (Mendeley) according to journal requirements and the citation style is consistent throughout the tex
  • some of the oldest references were updated; as permitted by journal guidelines only the most relevant ones were left and there is now only one reference left dating back to 20 years ago 
  • the "matherial and methods" paragraph has been aded

Reviewer 4 Report

Comments and Suggestions for Authors

This is an interesting review by Puca et al. The review is thorough and well organized in a logical manner. 

Two elements/factors that the authors may want to consider in terms of effect is diet and obesity status. There are several studies that indicate diet and particularly fiber content be related to outcomes relevant to IBS, while obesity status confers an inflammatory state. Regardless of the drug status the aforementioned conditions can constitute significant modulators of outcomes and it would thus be warranted to be discussed. While this is not the focus of the review it would make the paper much stronger if these parameters were considered in brief in the discussion at the end.

Comments on the Quality of English Language

English needs some improvement in certain parts of the manuscript to appear more professional in the narrative. 

Author Response

Thank you for your valid review and for the appreciation of our job. 

First, as you mentioned, we extensivley improved the quality of english writing in the text. 

Furthermore, since you and another reviewer underlined the possible role of nutrition and obesity in determining drug resistance, we added a dedicated small paragraph at the end of the main text . We hope you like this new part of the paper.